# Solving Engineering Optimization Problems Based on Multi-Strategy Particle Swarm Optimization Hybrid Dandelion Optimization Algorithm

**DOI:** 10.3390/biomimetics9050298

**Published:** 2024-05-17

**Authors:** Wenjie Tang, Li Cao, Yaodan Chen, Binhe Chen, Yinggao Yue

**Affiliations:** 1School of Intelligent Manufacturing and Electronic Engineering, Wenzhou University of Technology, Wenzhou 325035, China; 2Intelligent Information Systems Institute, Wenzhou University, Wenzhou 325035, China

**Keywords:** dandelion algorithm, particle swarm optimization algorithm, function optimization, multi-objective optimization, Levy flight

## Abstract

In recent years, swarm intelligence optimization methods have been increasingly applied in many fields such as mechanical design, microgrid scheduling, drone technology, neural network training, and multi-objective optimization. In this paper, a multi-strategy particle swarm optimization hybrid dandelion optimization algorithm (PSODO) is proposed, which is based on the problems of slow optimization speed and being easily susceptible to falling into local extremum in the optimization ability of the dandelion optimization algorithm. This hybrid algorithm makes the whole algorithm more diverse by introducing the strong global search ability of particle swarm optimization and the unique individual update rules of the dandelion algorithm (i.e., rising, falling and landing). The ascending and descending stages of dandelion also help to introduce more changes and explorations into the search space, thus better balancing the global and local search. The experimental results show that compared with other algorithms, the proposed PSODO algorithm greatly improves the global optimal value search ability, convergence speed and optimization speed. The effectiveness and feasibility of the PSODO algorithm are verified by solving 22 benchmark functions and three engineering design problems with different complexities in CEC 2005 and comparing it with other optimization algorithms.

## 1. Introduction

With the continuous development of science and technology, the complexity and scale of various practical application problems and optimization problems are increasing, and the traditional optimization problem-solving methods are no longer suitable for solving complex problems or make it difficult to meet the needs of high-precision solutions [1,2]. In recent years, swarm intelligence algorithms inspired by various biological groups in nature have been widely studied by international scholars, such as particle swarm optimization (PSO) [3], the butterfly optimization algorithm (BOA) [4], the SALP Swarm Algorithm (SALP) [5] and so on. This kind of algorithm has been widely used in solving optimization problems and other scientific fields because of its simple principle, high flexibility and high efficiency. Optimization algorithms provide a powerful tool for the engineering field, as they can accelerate the design process, improve system performance, reduce costs, optimize resource utilization and provide support for engineering decision-making [6,7]. Optimization algorithms have always been a hot topic in the field of engineering. With the development of society, more and more complex optimization problems have emerged. Traditional optimization algorithms can no longer meet requirements, and more efficient intelligent optimization algorithms are urgently needed. Therefore, people are focusing on swarm intelligence optimization algorithms, such as gray wolf optimization (GWO) [8], the Whale Optimization Algorithm (WOA) [9], Firefly FA (Firefly Algorithm) [10] and the Sparrow Search Algorithm (SSA) [11]. These algorithms are not limited in search space and objective function form and have good optimization ability, so they have been widely used in real life and engineering fields [12,13]. However, as the constraints of practical problems become more and more stringent, the applicability of many such algorithms has been insufficient, which requires the higher performance and higher applicability of optimization algorithms to deal with these complex application problems.

Dandelion Optimizer was first proposed by S. Zhao and T. Zhang in 2022, which was inspired by the seed propagation of dandelion plants and the fluffy structure of the flowers [14]. The DO algorithm is designed to solve engineering and scientific optimization problems, and its goal is to search the global optimal solution by simulating the growth and propagation mechanism of dandelion plants [15,16]. The algorithm has been successfully applied in engineering applications of artificial intelligence, antenna arrays, the design of independent microgrid systems and economic analyses [17]. However, due to the limitation of its own population, dandelion is still weak in global search ability, and it can easily fall into local optimum [18]. The long-distance flight process of dandelion seeds by wind was simulated, and two main factors, wind speed and weather, were added. Brownian motion and Levy flight were introduced to describe the trajectory of seeds.

In the original PSO and DO algorithms, there is a defect of easily falling into local optima when solving complex problems such as multidimensional problems and nonlinear optimization. In order to address these issues, this paper proposes a multi-strategy improved PSODO algorithm. The main contributions of our algorithm are as follows:Combining the particle-updating rules of the DO algorithm with the particle position and velocity-updating rules of the PSO algorithm, thereby expanding its search range in the solution space.Introducing a velocity decay strategy. In the early stages of algorithm optimization, higher velocities help particles explore the search space more widely to find the global optimal solution. However, as the iterations progress, excessively high velocities may cause the algorithm to skip some potentially excellent solutions. The velocity decay strategy is used to control the particle velocities in the population, providing a finer nutrient search for the later stages of the algorithm search process, thus balancing exploration and exploitation.Improving the stability of solutions. By mixing these two algorithms and reducing the learning rate of PSO, the stability of solutions can be further improved, reducing oscillation problems during the solving process.

The organization of the subsequent sections in this paper is as follows: Section 2 introduces the related works. Section 3 provides a detailed introduction to the dandelion optimization algorithm, including its rising, falling and landing phases. Section 4 elaborates on the PSODO algorithm, including the background and pseudocode of the PSO algorithm, the concept of combining PSO and DO algorithms, the process and the pseudocode of the PSODO algorithm. Section 5 presents the simulation results of the PSODO algorithm and six other algorithms, including optimal values, averages and variances of 22 benchmark functions; convergence curves of seven algorithms; the feasibility of the velocity decay strategy; and three typical engineering problems such as the three-bar truss design problem (TTD), pressure vessel design problem (PVD) and compression spring design problem (CSD). Finally, the paper concludes with conclusions and future prospects.

## 2. Related Works

The dandelion optimization algorithm is a heuristic algorithm based on dandelion falling reproduction. It is not easy for the algorithm itself to fall into local extremum and it has strong robustness. However, the falling operation of the algorithm adopts a fixed step size, which may wander around the optimal position and consume time, resulting in the loss of dandelion individuals with high fitness [19]. The dandelion optimization algorithm is a new optimization algorithm, which is based on the diffusion process of dandelions in nature and finds the optimal solution through population synergy [20]. Compared with other algorithms, the dandelion algorithm has the advantages of fast convergence and easy implementation [21,22]. Therefore, the dandelion algorithm has a wide application prospect in solving single-objective optimization problems. The dandelion optimization algorithm has been paid attention by a large number of researchers, and many improved algorithms have been proposed to solve complex engineering application problems. Erda ş et al. [23] proposed the application of the dandelion optimization algorithm to the optimization design and production process of seat brackets to optimize the production process, reduce costs and improve production efficiency. Wang et al. [24] proposed a multi-threshold segmentation method for breast cancer images based on an improved dandelion optimization algorithm, which achieved the highest fitness value and the fastest convergence speed when using the same threshold number. Han et al. [25] proposed an improved dandelion algorithm UAV path planning strategy. The algorithm proposes a new coding strategy, in which each dandelion represents a foothold of UAV, and the whole dandelion population is regarded as a whole deployment, which improves the efficiency of data collection.

However, after further study, the dandelion optimization algorithm does not have a high enough convergence accuracy, too easily falls into local optimum, and so on, for solving complex optimization problems. According to the theorem of “no free lunch”, there is no algorithm suitable for solving any optimization problem. Therefore, it is very meaningful to improve the algorithm from different angles. Relevant scholars have made some improvements in view of its shortcomings. For example, in reference [25] proposed a dandelion optimization algorithm based on insanity self-adaptation, which uses a chaotic map to initialize the population to improve individual diversity and introduces an insanity operator to update the leader position to improve the development ability of the algorithm. Reference [26], differential mutation operation is introduced into the leader position update mechanism of the standard dandelion optimization algorithm, which improves the global optimization ability of the algorithm and accelerates the convergence speed of the algorithm. Khalil et al. [27] proposed a new cascaded loop controller based on the dandelion optimization algorithm, which combines fractional proportional derivatives with filters and fractional proportional skewed integral derivatives (FPDN-FPTID) to improve the LFC of single region and multi region IMGs. Reference [28] proposed a dandelion optimization algorithm based on an adaptive normal cloud model, which uses a normal cloud model mechanism to help the algorithm jump out of local optimum and improve the diversity and search accuracy of the algorithm.

In 1995, Dr. Eberhart and Dr. Kennedy proposed the particle swarm optimization algorithm (PSO), which was inspired by birds’ foraging behavior. Compared with other algorithms, these two algorithms are easy to implement, simple in principle and strong in optimization ability. However, with the further development of research, it is found that there are still some shortcomings in the application process, but these two algorithms provide improved ideas for later researchers. To solve the above problems, this paper proposes a multi-strategy particle swarm hybrid dandelion optimization algorithm (PSODO). The algorithm first uses particle swarm optimization to update the population in the initial stage of population, then calculates the fitness value of each particle and dandelion, that is, the objective function value, then updates the individual and global optimum, updates the individual optimal solution (*p_Best_*) for each particle and dandelion and then updates the global optimal solution (*g_best_*) according to the individual optimal solution (*p_Best_*). It is compared with the PSO, GWO, WDO, SCA, TSA and DO algorithms. The effectiveness and feasibility of PSODO are verified by solving 22 benchmark functions and three engineering design cases of CEC2005 and comparing it with other optimization algorithms.

## 3. Dandelion Optimization Algorithm

Dandelion optimizer (DO) was proposed by Shijie Zhao et al. in 2022, which simulates the long-distance flight of dandelion seeds by wind and includes three stages, namely, the ascending stage, descending stage and landing stage [29]. The algorithm considers two main factors, wind speed and weather, and introduces Brownian motion and Levy flight to describe the trajectory of seeds [30]. In the ascending stage, according to different weather conditions, seeds move in a spiral ascending way in the community or flow locally [31]. In the descent stage, by constantly adjusting the flight direction in the global space, the seeds descend steadily after rising to a certain height; in the landing phase, under the influence of wind and weather, seeds randomly select positions to land. Dandelion seed transmission experiences three stages and realizes population evolution [32].

The framework of the DO algorithm roughly consists of four parts: population initialization, population fitness calculation, population updating and global optimization selection. It contains two main parameters, namely, seed propagation radius α and local search coefficient K. Seed propagation radius α and local search coefficient K change with time in the iterative process; α is used to adjust the global search step size and local search coefficient K is used to adjust the process from exploration to development and avoid local optimization by random numbers obeying normal distribution [33].

A dandelion population matrix seed with *N* seeds and *d*-dimensional decision space is defined, and its *i*-th seed can be expressed as $X_{i}=[X_{i}^{1},\;X_{i}^{2},\;X_{i}^{3},\;\ldots ,\;X_{i}^{d}]$, *i* = 1, 2, …, *N*. The population is initialized as Formula (1):(1)$$X_{i}=L_{B}+r_{i}\times (U_{B}-L_{B})$$

Among them, parameter *r_i_* is a random number between (0, 1) which obeys normal distribution, *U_B_* is the maximum value in decision space, *L_B_* is the minimum value in decision space, *N* is the maximum row value of the population matrix, *d* is the maximum column value of the population matrix, and the same characters in the following are synonymous [34].

(1)Rising stage

Dandelion seeds disperse after reaching a certain height. Affected by wind speed, temperature and humidity, the rising height of dandelion seeds is different, which can be divided into two situations according to the weather. On a sunny day, the wind speed obeys the normal distribution of logarithmic property ln*Y* ~ *N* (μ, σ^2^) randomly and uniformly distributed along the Y axis. Dandelion seeds have more opportunities to spread to distant areas. In this process, the DO algorithm emphasizes exploration. In the exploration space, dandelion seeds are randomly blown to different positions by the wind. Wind speed determines the flying height of seeds. The stronger the wind, the farther the seeds are scattered. The wind speed constantly adjusts the vortex above the seed, and the rising posture is spiral. In this process, the expression corresponding to the seed ascending stage is shown in Equation (2) [35].
(2)$$X_{t+1}=X_{t}+\alpha v_{x}v_{y}\ln Y(X_{s}-X_{t})$$
where *X_t_* is the position of the seed in *t* iterations. *X_s_* is the randomly selected location in the search space during the iteration. The random locations selected by *X_s_* are shown in Equation (3).
(3)$$X_{s}=rand(1)(U_{B}-L_{B})+L_{B}$$
where *U_B_* and *L_B_* values are set to 1 and 0, respectively, and *rand*(1) is a random number. The function ln*Y* obeys the lognormal distribution of *μ* = 0 and σ^2^ = 1, and its formula is shown in (4).
(4)$$\ln Y=\left\{\begin{array}{ll} \frac{1}{y\sqrt{2\pi }}\exp[-\frac{1}{2\sigma^{2}}{\left(\ln y\right)}^{2}] & ,\;y\geq 0\\ 0, & \;,\;y<0 \end{array}\right.$$

Parameter *Y* is the standard normal distribution *N* (0, 1). The search step *α* is adjusted as an adaptive parameter, as shown in Equation (5).
(5)$$\alpha =rand()(\frac{1}{T^{2}}t^{2}-\frac{2}{T}t+1)$$

Random perturbation factor *α* is in [0, 1]. With the increase in iteration times, *α* nonlinearity decreases and tends to 0. In the initial stage of the algorithm, the disturbance is relatively large, and the algorithm pays more attention to the global search. In the later stage of the algorithm, *α* decreases and the algorithm turns to the local search. Turning to a local search after a global search is more conducive to accurate convergence. When the separation vortex acts on dandelion, the lift component coefficients *v_x_* and *v_y_* will be produced. The variable dimensional force formula is shown in Equation (6).
(6)$$\left\{\begin{array}{l} r=e^{-\theta }\\ v_{x}=r\cos\theta \\ v_{y}=r\sin\theta \end{array}\right.$$

The parameter *θ* is a random number [−π, π]. On rainy days, affected by air humidity, air resistance and other factors, dandelion seeds cannot fully rise with the wind, and seeds are developed in local areas. The corresponding formula is shown in Equation (7).
(7)$$X_{t+1}=X_{t}k$$

The parameter *k* value adjusts the local search area of dandelion. The *k* value is calculated as shown in Equation (8).
(8)$$\left\{\begin{array}{l} q=\frac{1}{T^{2}-2T+1}t^{2}-\frac{1}{T^{2}-2T+1}t+\frac{1}{T^{2}-2T+1}+1\\ k=1-rand()q \end{array}\right.$$

In Equation (8), the value of *k* oscillates convexly downward. This is beneficial for the algorithm to search the global region with a long step in the initial stage and develop the local region with a short step in the later stage. With the increase in iteration times, the parameter *k* value is closer and closer to 1, which can ensure that the population finally converges to the optimal search individual. Based on the above analysis, the analytical formula describing the ascending stage of dandelion seeds is shown in Formula (9).
(9)$$X_{t+1}=\left\{\begin{array}{ll} X_{t}+\alpha v_{x}v_{y}\ln Y(X_{s}-X_{t}) & ,rand(n)<1.5\\ X_{t}k & ,\mathrm{others} \end{array}\right.$$
where *rand*(*n*) is a random number that obeys the standard normal distribution. Equation (9) gives the approximate position of dandelion seed evolution: under sunny weather, the position information to update is randomly selected, the spiral movement direction of seeds is corrected by vx and vy and the exploration process is emphasized. On rainy days, dandelion seeds are developed in local areas. Using the exploration and development of the normal distribution dynamic control algorithm of random numbers, the truncation point value is set to 1.5, which makes the algorithm more global-oriented and traverses the whole search space as much as possible in the initial stage, providing a correct direction for the next iterative optimization.

(2)Descending stage

In this stage, the DO algorithm emphasizes the search and optimization of the algorithm. Dandelion seeds descend steadily after rising to a certain height. The DO algorithm uses Brownian motion obeying normal distribution to simulate the movement track of dandelion seeds, so that individuals can easily traverse more search areas in the iterative process. Average position information after the rising stage reflects the stability of dandelion descent, which is helpful for the whole population to search and develop the optimal individual areas, as shown in Formula (10).
(10)$$X_{t+1}=X_{t}-\alpha \beta_{t}(X_{t}^{mean}-\alpha \beta_{t}X_{t})$$

Parameter *β_t_* is a random number and obeys Brownian motion with normal distribution. $X_{t}^{mean}$ is the average position information of the population at the *t*-th iteration, as shown in Equation (11).
(11)$$X_{t}^{mean}=\frac{1}{N}\sum_{i=1}^{N_{p}}X_{i}$$

Parameter *N_p_* is the number of populations. The average position information of the population determines the evolution direction of individuals, which is very important for the iterative update of individuals. Irregular Brownian motion based on a global search can make individuals escape from the local extremum with high probability when iteratively updating and force the population to search for optimization in the range close to global optimum.

(3)Landing phase

The DO algorithm focuses on development. After the first two stages, dandelion seeds randomly select positions to land. With the increase in iteration times, the algorithm is expected to converge to the global optimal solution, reflecting the approximate position where seeds are most likely to survive. Therefore, after obtaining the approximate position where the optimal dandelion seeds are most likely to grow, the algorithm can accurately converge to the global optimal solution by using the local information of the current elite individuals. With the continuous evolution of the population, the final algorithm converges to the global optimal solution, and the expression is Formula (12).
(12)$$X_{t+1}=X_{elite}+Levy(\lambda )\alpha (X_{elite}-X_{t}\delta )$$

Parameter *X_elite_* is the optimal position of the seed generated in the *t*-th iteration. Levy (*λ*) is a Levy flight function, which is used to enhance the local search ability. Parameter *δ* is a linearly increasing function, and its value is [0, 2], which is used to avoid over-exploitation and make it converge to the global optimum accurately, and *T* is the total number of iterations.
(13)$$Levy(\lambda )=s\omega \sigma /{\left|t\right|}^{\frac{1}{\beta }}$$
(14)δ=2t/T

Parameter *β* is a random number, and its value range is [0, 2], and 1.5 is taken in this paper. Parameter *s* is 0.01. Parameters *w* and *t* are random numbers with values in [0, 1].

## 4. Dandelion Algorithm Optimized by PSO (PSODO) Particle Swarm Optimization

Inspired by the regularity of bird foraging behavior, James Kennedy and Russell Eberhart established a simplified algorithm model, which, after years of improvement, ultimately formed the particle swarm optimization (PSO) algorithm [36]. The idea of particle swarm optimization (PSO) originates from research on the foraging behavior of birds. Birds find the best destination through collective information sharing. For example, imagine such a scenario: birds randomly search for food in the forest, and they want to find the position with the largest amount of food or there is only one piece of food in a certain area (that is, the optimal solution studied in the optimization problem), but all birds do not know where the food is and can only feel the approximate direction of the food [37,38]. Therefore, each bird searches along the direction determined by itself and records the position where it has found the most food in the process of searching. At the same time, all birds share the position and amount of food found every time, so that each bird in the flock can judge whether it has found the most food (optimal solution). At the same time, the information of the optimal solution is transmitted to the whole bird flock, and finally, the whole bird flock can gather around the food source, that is, we find the optimal solution, that is, the problem converges [39]. Particle swarm optimization (PSO) has the advantages of fast convergence, few parameters, a simple algorithm and easy implementation (for high-dimensional optimization problems, it converges to the optimal solution faster than a genetic algorithm), but it also has the problem of falling into the local optimal solution, so it depends on good initialization [40].

The PSO algorithm updates rules. First, the expression (15) for updating the velocity and position of each particle in PSO is as follows:(15)$$V_{ij}=\omega \cdot V_{ij}+c_{1}\cdot r_{1}\cdot (P_{ij}-X_{ij})+c_{2}\cdot r_{2}\cdot (G_{j}-X_{ij})$$

Parameter *V_ij_* is the velocity of particle *i* in dimension *j*; *X_ij_* is the position of the particle in dimension *j*; parameter *P_ij_* is the individual optimal position of the particle, which is the global optimal position; ω is the inertia weight, which controls the influence of the previous velocity on the current velocity to keep its motion inertia and has the ability to search for new areas; *c*_1_ and *c*_2_ are acceleration factors; *r*_1_ and *r*_2_ are random numbers; *r*_1_ and *r*_2_ are the rand() function, and their value range is [0, 1). The location update Formula (16) is as follows:(16)$$X_{ij}=X_{ij}+V_{ij}$$

The pseudo code of the PSO algorithm is shown in Algorithm 1.
**Algorithm 1: Particle swarm optimization****Input:** *POP*, *Dim*, *T*, [*lb*, *ub*], [*v_min_*, *v_max_*].**Output:** *Best_fitness*, *Best_pos*.1: Initialize population with random positions and velocities within the search space [*lb*, *ub*].2: Evaluate fitness for each particle 3: Update personal best position (*p_Best_Pos_*) and personal best fitness (*p_Best_fitness_*) if necessary.4: Determine if the maximum *T* has been reached.5: Update (*p_Best_Pos_*) and *p_Best_fitness_* based on the best particle’s position and the fitness. Update based on the position and fitness of the best particles (*p_Best_Pos_*) and *p_Best_fitness_*.6: Record the best fitness value in the iteration.7: Update the inertia weight ω if necessary.8: Reached cycle count *T*, ending cycle.9: Return *p_Best_Pos_* and *p_Best_fitness_*.

### Hybrid Strategy of PSODO Algorithm

The PSODO algorithm combines the characteristics of particle swarm optimization (PSO) and dandelion optimization (DO), and realizes mixing through their updating rules. In the PSO part, we use the speed update formula of PSO, including inertia weight and individual and social learning factors, and guide the particle search through individual optimal position (*p_Best_*) and global optimal position (*g_Best_*). In the part of DO, we use the randomness of the DO algorithm to increase the diversity of the search through random numbers and different update strategies and use the location update rules of DO, including a random selection update strategy, dandelion algorithm adjustment and so on. More randomness is introduced through random numbers, including the random selection and updating strategy of the DO part and dandelion optimization parameters, etc. This randomness helps the algorithm jump out of the local optimal solution and increase the global search ability. In addition, we also introduce a deceleration strategy to balance the global search and local search. By properly adjusting the deceleration strategy, we can introduce particle swarm optimization in the early stage of the algorithm to make particles move in the whole search space at a higher speed, realize the global search and hopefully find the global optimal solution. In the later stage, the velocity of particles is gradually reduced, so that particles can search more finely near the global optimal solution and improve the accuracy of the solution. The flow chart of the PSODO algorithm is shown in Figure 1.

The pseudocode of the PSODO algorithm is shown in Algorithm 2.
**Algorithm 2: PSODO algorithm****Input:** *POP*, *Dim*, *T*, [*lb*, *ub*], [*v_min_*, *v_max_*].**Output:** Best Individual *X_elite_*. Step 1: Define a dandelion population matrix seed with N seeds and d dimension decision space, and its i-th seed can be expressed as $X_{i}=[X_{i}^{1},\;X_{i}^{2},\;X_{i}^{3},\;\ldots ,\;X_{i}^{d}]$, *i* = 1, 2, …, *N*. Initialize the population as Formula (1). DO algorithm initializes settings. Step 2: Calculate the initial seed position and select the best position of dandelion seeds. Step 3: Evaluating the seed of the population and selecting the best individual of the population. Step 4: Determining global search or local development through a random number *r* obeying normal distribution: If *r* < 0.7, rising when the weather is clear, the position of the seed at this time is Formula (2), and in order to achieve the global search, the speed and position of the seed are updated (15). If *r* > 0.7, when the weather is rainy, the position of the seed at this time is Formula (7), and in order to improve the local search ability, the position is updated (16). Step 5: After rising for a certain distance, the algorithm still focuses on exploration at this time, and the seed begins to gradually descend according to Brownian motion, and the seed position is shown in Formula (10). Step 6: Dandelion begins to land at this time, and randomly selects the landing place on land according to Levy flight, and the algorithm begins to converge and enter the development stage. The seed position is expressed as Formula (12). Step 7: Update the best individual *X_elite_*. Step 8: If the iteration is not finished, return to step 2, otherwise output *X_elite_*. Step 9: Output the optimal result

## 5. Simulation Experiment Results and Analysis

### 5.1. Experimental Environment and Test Function

The PSODO algorithm is implemented based on MATLAB language. The experimental environment is configured as the Windows 11 operating system, CPU AMD Ryzen 97945HX, and the MatlabR2022a experimental platform is used for simulation. In order to validate the performance advantages of the PSODO algorithm, the benchmark functions are used to simulate the standard particle swarm optimization (PSO) [41], dandelion optimization (DO) [42], gray wolf optimization (GWO) [43], sine–cosine optimization (SCA) [44], wind-driven optimization (WDO) [45] and tree species optimization (TSA) [46]. The test functions are all selected from the global test function library of CEC 2005, and their function forms, search ranges and optimal solutions are shown in Table 1.

### 5.2. Comparison of Test Function Results

In order to verify the effectiveness of the hybrid PSODO algorithm, the above algorithms are run 30 times on F1–F22 functions, the population is set to 50, the dimensions are set to 30/60 (some functions adopt fixed dimensions) and the maximum iteration times are 500 times. The average value (Mean), standard deviation (Std), optimal value (Best) and Worst value (Worst) of each algorithm are taken as performance indicators, and the results are shown in Table 2 below, where bold indicates the optimal result.

It can be seen from Table 2 that in 30 dimensions, PSODO obtains the optimal values on functions F2, F3, F4 and F7–F10, especially on F4, F7 and F9, while all algorithms do not converge to the optimal values on functions F5 and F6. The optimal value is also found on the multimodal function F8–F10. On the fixed dimension functions F13–F22, PSODO finds the optimal value. It can be seen from Table 3 that PSODO can still achieve optimal values on functions F2, F3, F4, F7–F10 and F13–F22 when the dimension is 60. To sum up, PSODO has good optimization performance in unimodal, multi-modal and fixed-dimensional functions and low-dimensional and high-latitude problems. In view of the poor performance of some functions, we introduce speed decline strategies to improve the optimization performance of the algorithm, including but not limited to a linear decline strategy and exponential decline strategy.

In order to compare the convergence of each algorithm on different functions more intuitively, this paper draws the convergence curves of the PSODO algorithm and the other six algorithms. Comparisons of the convergence curves of seven algorithms are shown in Figure 2, with the horizontal axis representing iteration and the vertical axis representing fitness values. As can be seen from Figure 2, the PSODO algorithm has the fastest convergence speed and the highest convergence accuracy on functions F5, F8, F13–F18 and F21 and can find nearly the optimal value at the beginning, especially functions F8 and F14. It converges to the optimal value in the form of an approximate straight line. For function F7, the convergence speed of the PSODO algorithm is slightly slower than that of GWO, ranking second. For functions F1–F4 and F9–F12, the convergence speed of the PSODO algorithm is slightly slower than that of GWO and WDO, ranking third. For functions F6, F19, F20 and F22, the convergence speed of the PSODO algorithm is relatively average, but the optimal value can be found in the end. Thus, it is proved that the performance of the PSODO algorithm is relatively good and the multi-strategy improvement–decreasing speed strategy is feasible and effective for improving the convergence speed and accuracy of the algorithm.

### 5.3. Engineering Problem Application

In order to verify the optimization performance of the PSODO algorithm in engineering applications, seven algorithms are applied to two common engineering problems: the three-bar truss design problem (TTD) and the pressure vessel design problem (PVD). They are run 30 times on each engineering problem, and the optimal value they find is recorded. Among them, the maximum number of iterations is 300, the population size is 30 and the penalty function is used to deal with inequality constraints.

#### 5.3.1. Three-Bar Truss Design Problem (TTD)

The three-bar truss design problem is a common structural form, which is widely used in bridges, buildings, mechanical equipment and other fields. The design optimization of bar-honing frames refers to adjusting the parameters such as the size, shape and connection mode of bars, so that the structure has the best performance and economy under certain constraints. The objectives of three-bar split frame design optimization mainly include the following aspects: (1) Structural strength and stiffness: The design of a three-bar split frame needs to meet certain strength and stiffness requirements to ensure that the structure will not be unstable or damaged during use. Optimal design can make the structure have the best strength and stiffness under external loads by adjusting the cross-sectional area and length of members. (2) Structure weight: The weight of the three-bar bin frame directly affects the cost of the structure and the convenience of transportation and installation. Optimal design can reduce the weight of members, so that the structure has the lightest weight on the premise of meeting the requirements of strength and stiffness. (3) Structural stability: The three-bar split frame needs to be kept stable when subjected to external loads to avoid instability and plastic deformation. Optimal design can make the structure have the best stability under external loads by adjusting the size and shape of members. (4) Economy of structure: The design of a three-bar split frame needs to consider the factors such as material cost, manufacturing cost and maintenance cost, so as to make the structure have the lowest total cost on the premise of meeting the performance requirements.

The design problem of a three-bar truss is the adjustment of the cross-sectional areas (*x*_1_ and *x*_2_) to solve the minimum volume of the three-bar truss under the constraint of the stress (*σ*) that each truss member can bear. Its mathematical model is as follows:(17)$$minf(x)=(2\sqrt{2x_{1}}+2)\times l$$
(18)$$g_{1}(\vec{x})=\frac{\sqrt{2x_{1}}+x_{2}}{\sqrt{2}x_{1}^{2}+2x_{1}x_{2}}P-\sigma \leq 0$$
(19)$$g_{2}(\vec{x})=\frac{x_{2}}{\sqrt{2}x_{1}^{2}+2x_{1}x_{2}}P-\sigma \leq 0$$
(20)$$g_{3}(\vec{x})=\frac{1}{\sqrt{2}x_{2}+x_{1}}P-\sigma \leq 0$$

$l=100\;\mathrm{cm};P=2\;{\mathrm{kN}/\mathrm{cm}}^{2};\sigma =2\;{\mathrm{kN}/\mathrm{cm}}^{2}$, $0\leq x_{i}\leq 1,\;i=1,2.$ From Table 4, the optimal parameters of PSODO are (0.78117, 0.4299) and the minimum volume is 263.8959. The PSODO algorithm is applied to the three-bar truss problem, and the experimental results are compared with several improved algorithms in other studies. Table 5 shows the optimal solutions obtained by different algorithms and the values of related decision variables. The results of three-bar truss design problems with seven algorithms are shown in Table 4. The convergence curve of the three-bar diffraction frame design problem is shown in Figure 3. The results show that the MEO algorithm is more competitive than other algorithms.

From the optimization results of seven algorithms in Table 4 and Table 5 and Figure 3, it is obvious that the PSODO algorithm can find better control parameter values and objective function values. Generally speaking, when the values of parameters *x*_1_ and *x*_2_ are 0.7886 and 0.4082, respectively, the volume of the three-bar diffraction reaches the minimum value of 263.8959. The optimization results further show that the PSODO algorithm has high optimization efficiency in solving three-bar truss problems.

#### 5.3.2. Pressure Vessels Design Problem

The research of pressure vessel design is an important and complex engineering field, which involves many key problems and challenges. The following are several common research directions: (1) Structural optimization: In the process of pressure vessel design, structural optimization is a key research direction. It includes the consideration of the mechanical properties of the material and the geometry of the vessel to minimize weight or cost while meeting specified strength and stiffness requirements. (2) Material selection: Material selection is another important research direction. Different materials have different physical and chemical properties, which have an important impact on the performance and reliability of pressure vessels. Researchers need to consider the strength, corrosion resistance, heat resistance and other characteristics of materials to choose the most suitable materials. (3) Fatigue life prediction 1: Because pressure vessels are subjected to cyclic loads during use, fatigue life prediction is a key problem. Researchers need to consider the fatigue strength of materials and the stress of containers, use related fatigue analysis methods to predict the life of containers, and take corresponding measures to prolong their service life. (4) Safety analysis: The safety of pressure vessels is an important research direction. Researchers need to analyze and evaluate the stress state, stress distribution and deformation of the container to ensure the safe operation of the container under different working conditions and find and prevent potential safety hazards in time.

The goal of pressure vessel design (PVD) is to minimize the total cost *f*(*x*) while meeting the production needs. Its four design variables are shell thickness *Ts*(*x*_3_), head thickness *T_h_*(*x*_4_), inner radius *R*(*x*_1_) and container length *L*(*x*_2_), where *Ts* and *T_h_* are integer multiples of 0.625, and *R* and *L* are continuous variables. The specific mathematical model is shown in Equations (21)–(26).

Objective function:(21)$$\min f(x)=0.6224x_{1}x_{3}x_{4}+1.7781x_{2}x_{3}^{2}+3.1661x_{1}^{2}x_{4}+19.84x_{1}^{2}x_{3}$$

Constraints:(22)$$g_{1}(x)=-x_{1}+0.0193x_{3}\leq 0$$
(23)$$g_{2}(x)=-x_{2}+0.00954x_{3}\leq 0$$
(24)$$g_{3}(x)=-\pi x_{3}^{2}x_{4}-\frac{4}{3}\pi x_{3}^{2}+1296000\leq 0$$
(25)$$g_{4}(x)=x_{4}-240\leq 0$$

Boundary constraints:(26)$$0\leq x_{1}\leq 99,\;0\leq x_{2}\leq 99,\;10\leq x_{3}\leq 200,\;10\leq x_{4}\leq 200$$

The PSODO algorithm is used to optimize the four key variables of the pressure vessel problem and obtain the optimal values, and the results are compared with the data of six algorithms that have solved the problem. Table 6 shows the values of the lowest cost and related variables obtained by each algorithm. The results of pressure vessel design problems of seven algorithms are shown in Table 7. The convergence curve of the pressure vessel design problem is shown in Figure 4. The results show that the PSODO algorithm is more competitive than other algorithms.

It can be seen from the data in Table 6 and Table 7 and Figure 4 that compared with other algorithms, the PSODO algorithm has a better optimization effect and saves engineering design costs. When the values of four core parameters, *T_s_*, *T_h_*, *R* and *L*, are 13.2089, 7.4947, 42.0984 and 176.6366, respectively, the lowest cost of the algorithm is 5885.3328.

#### 5.3.3. Compression Spring Design Problem

The goal of compression spring design (CSD) is to minimize its mass *f*(*x*) under full constraints, which includes four inequality constraints, namely, maximum deflection, shear stress, oscillation frequency and outer diameter limit, and three design variables, which are the average diameter *d* of the spring coil, the diameter *d* of the spring wire and the effective number *N* of the spring. The tension/compression spring problem is a classical structural engineering design problem, which aims to minimize the weight of the spring by satisfying four constraints: deflection, shear stress, fluctuation frequency and outer diameter. The specific mathematical model is shown in Equations (27)–(32).

Objective function:(27)$$\min f(x)=(N+2)Dd^{2}$$

Constraints:(28)$$g_{1}(x)=1-\frac{D^{3}N}{{71785d}^{4}}\leq 0$$
(29)$$g_{2}(x)=\frac{4D^{2}-\mathrm{dD}}{12566({\mathrm{Dd}}^{3}-d^{4})}+\frac{1}{5108d^{2}}-1\leq 0$$
(30)$$g_{3}(x)=1-\frac{140.45d}{D^{2}N}\leq 0$$
(31)$$g_{4}(x)=\frac{D+d}{1.5}-1\leq 0$$

Boundary constraints:(32)$$0.05\leq x_{1}\leq 2,\;0.25\leq x_{2}\leq 1.3,\;2\leq x_{3}\leq 15$$

Based on the improved algorithm PSODO, the tension/compression spring problem is optimized and the values of related parameters are obtained. The optimized results are compared with algorithms in other articles. Detailed information is shown in Table 8. The convergence curve of the compression spring design problem is shown in Figure 5.

It can be seen from the results in Table 8 and Figure 5 that when the values of three parameters, *d*, *D* and *N*, are 0.051, 0.3175 and 14.0183, respectively, the spring weight obtained by the PSODO algorithm reaches the optimal value of 0.01271. Generally speaking, the PSODO algorithm has better performance than the original DO algorithm and other meta-heuristic algorithms in dealing with compression spring problems.

In this paper, the PSODO algorithm is applied to two engineering problems with different complexities. The complexity range of these problems varies from two design variables to four design variables. By dealing with complex problems with different numbers of parameters, the performance of the PSODO algorithm in solving engineering problems is further demonstrated. The design scheme given by the PSODO algorithm is compared with the scheme proposed by the existing algorithms in the literature. The comparison results show that the design cost of the scheme proposed by the PSODO algorithm is far lower than that of the original DO algorithm and other comparison algorithms, and it is an efficient algorithm to solve engineering optimization problems. At the same time, the short running time of the PSODO algorithm also shows the practicability and stability of the improved algorithm in engineering optimization problems with multiple complex constraints.

## 6. Conclusions

In this paper, based on the original PSO algorithm and DO algorithm, a multi-strategy hybrid dandelion optimization algorithm is proposed. According to the characteristics of particle swarm optimization and the dandelion population, the proposed algorithm is improved, and the speed-decreasing strategy is introduced to improve the performance of the algorithm. The PSODO algorithm is compared with other six optimization algorithms on 22 benchmark functions in multi-dimensional experiments, and the convergence curves of each algorithm are analyzed, which fully proves the effectiveness of the improved strategy. Finally, PSODO is applied to three complex engineering optimization problems, which proves that PSODO has good engineering practice value and good popularization value.

Although the PSODO algorithm has great advantages in data processing, it still has some shortcomings. Firstly, the algorithm adopts an optimization strategy, which increases the complexity of the algorithm. Secondly, this algorithm still lacks a large number of experiments on other difficult data-processing tasks, and this article only applies this algorithm to test some engineering application problems and benchmark function testing. Therefore, we will continue to apply the PSODO algorithm to other practical problems such as feature selection, image segmentation, parameter optimization and processing in the future, and further discuss the optimization performance of the proposed algorithm.

## Figures and Tables

**Figure 1 biomimetics-09-00298-f001:**
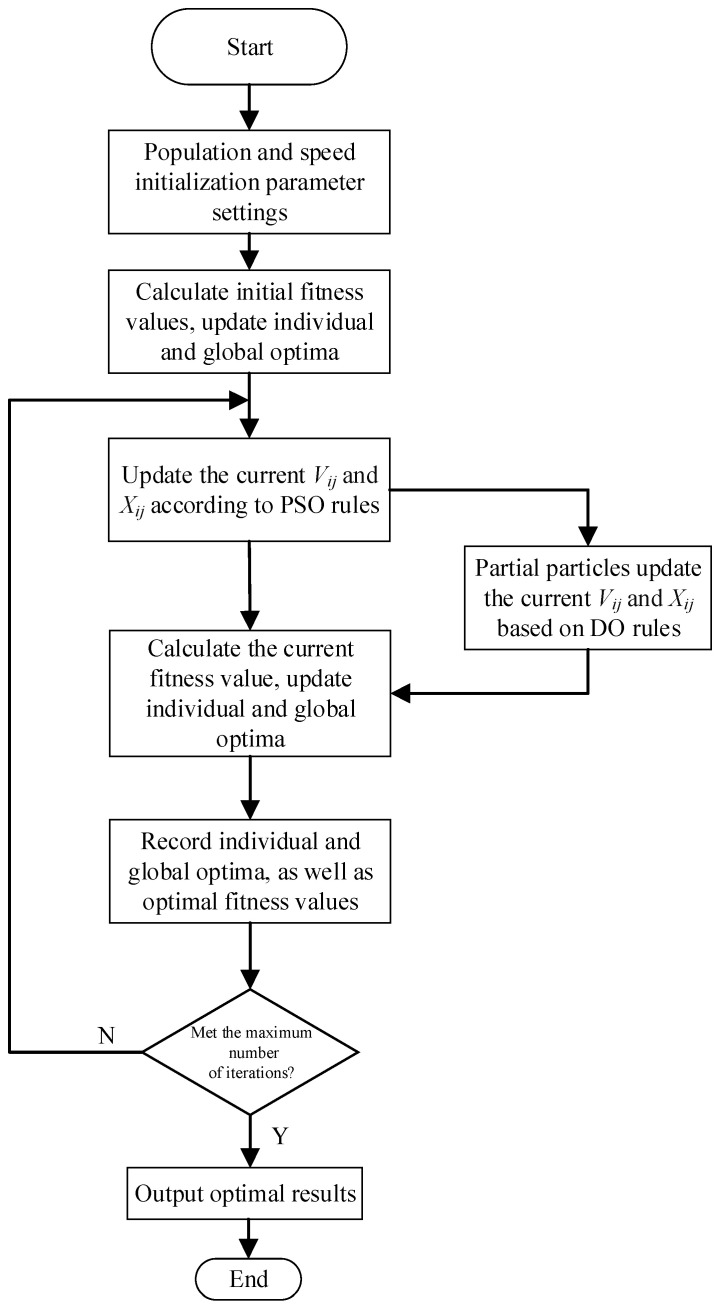
The flow chart of the PSODO algorithm.

**Figure 2 biomimetics-09-00298-f002:**
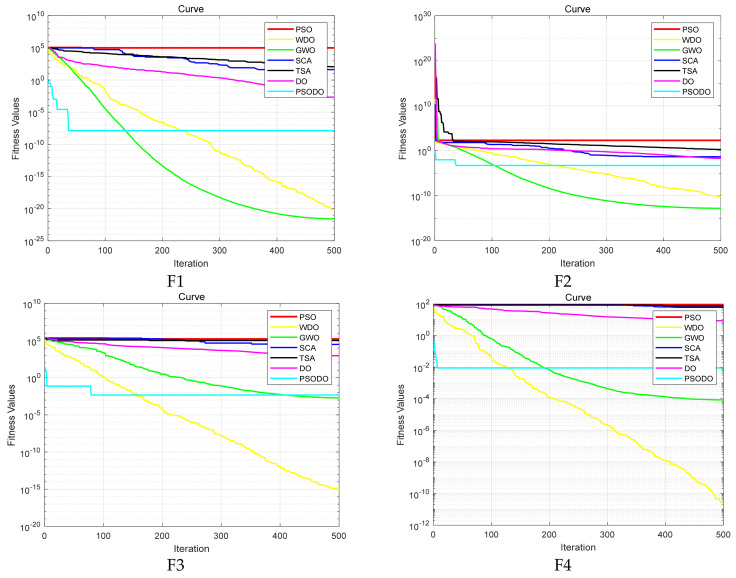
Comparison of the convergence curves of seven algorithms.

**Figure 3 biomimetics-09-00298-f003:**
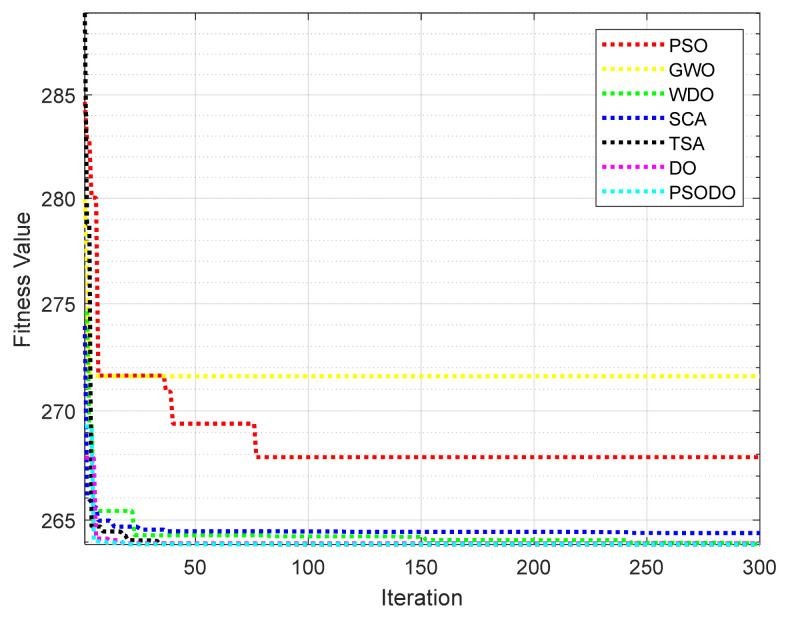
Convergence curve of the three-bar truss design problem.

**Figure 4 biomimetics-09-00298-f004:**
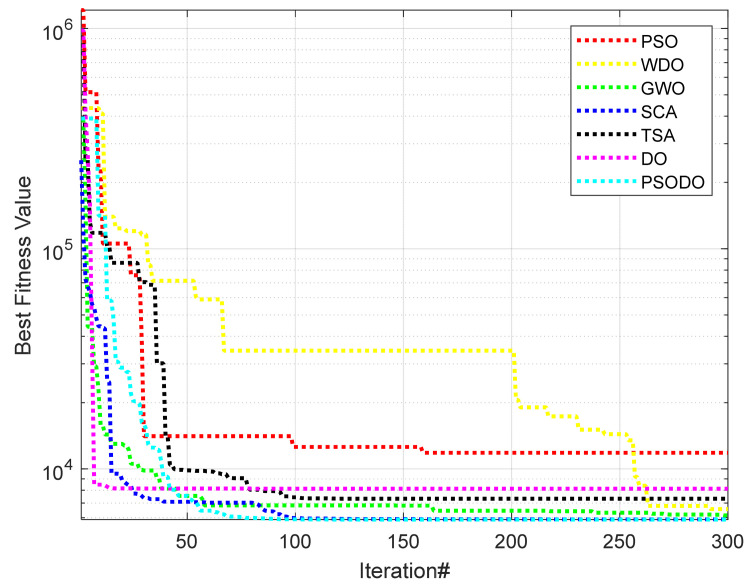
Convergence curve for pressure vessel design problems.

**Figure 5 biomimetics-09-00298-f005:**
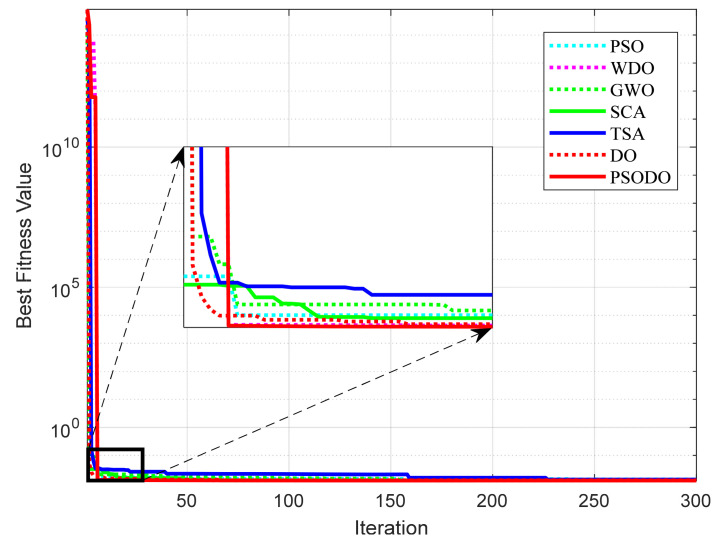
Convergence curve of compression spring design problem.

**Table 1 biomimetics-09-00298-t001:** Test functions.

Function	Equation	Dimension	Bounds	Optimum
F1	$\sum_{i=1}^{n}x_{i}^{2}$	30	[−100, 100]	0
F2	$\sum_{i=1}^{n}\left|x_{i}\right|+\prod_{i=1}^{n}\left|x_{i}\right|$	30	[−10, 10]	0
F3	${\sum_{i=1}^{n}\left(\sum_{j=1}^{i}x_{j}\right)}^{2}$	30	[−100, 100]	0
F4	$\underset{i}{\max}\left.\left\{\left|x_{i}\right|,1\leq i\leq n\right.\right\}$	30	[−100, 100]	0
F5	$\sum_{i=1}^{n-1}\left[100{\left(x_{i+1}-x_{i}^{2}\right)}^{2}+{\left(x_{i}-1\right)}^{2}\right]$	30	[−30, 30]	0
F6	$\sum_{i=1}^{n}{\left(\left[x_{i}+0.5\right]\right)}^{2}$	30	[−100, 100]	0
F7	$\sum_{i=1}^{n}ix_{i}^{4}+random[0,1)$	30	[−1.28, 1.28]	0
F8	$\sum_{i=1}^{n}\left[x_{i}^{2}-10\cos(2\pi x_{i})+10\right]$	30	[−5.12, 5.12]	0
F9	$\begin{array}{l} -20\exp\left(-0.2\sqrt{\frac{1}{n}\sum_{i=1}^{n}x_{i}^{2}}\right)-\\ \exp\left(\frac{1}{n}\sum_{i=1}^{n}\cos\left(2\pi x_{i}\right)\right)+20+e \end{array}$	30	[−50, 50]	0
F10	$\frac{1}{4000}\sum_{i=1}^{n}x_{i}^{2}-\prod_{i=1}^{n}\cos\left(\frac{x_{i}}{\sqrt{i}}\right)+1$	30	[−600, 600]	0
F11	$\begin{array}{l} \frac{n}{\pi }\{10\sin\left(\pi y_{1}\right)+\sum_{i=1}^{n-1}{\left(y_{i}-1\right)}^{2}[1+10\sin^{2}\left(\pi y_{i+1}\right)]+{\left(y_{n}-1\right)}^{2}\}\\ +\sum_{i=1}^{n-1}u\left(x_{i},10,100,4\right)\\ y_{i}=1+\frac{1}{4}(x_{i}+1)\\ u(x_{i},a,k,m)=\left\{\begin{array}{l} k{\left(x_{i}-a\right)}^{m},x_{i}>a,\\ 0,-a\leq x_{i}\leq a,\\ k{\left(-x_{i}-a\right)}^{m},x_{i}<-a. \end{array}\right. \end{array}$	30	[−50, 50]	0
F12	$\begin{array}{l} 0.1\{\sin^{2}(3\pi x_{1})+\sum_{i=1}^{n-1}u{\left(x_{i}-1\right)}^{2}[1+\sin^{2}(3\pi x_{i+1})]+\\ {\left(x_{n}-1\right)}^{2}[1+\sin^{2}(2\pi x_{n})]\}+\sum_{i=1}^{n}u(x_{i},5,100,4) \end{array}$	30	[−50, 50]	0
F13	$\left[\frac{1}{1500}+\sum_{j=1}^{50}\frac{1}{j+\sum_{i=1}^{30}{\left(x_{i}-a_{ij}\right)}^{6}}\right]$	2	[−65.536, 65.536]	1
F14	$\sum_{i=1}^{11}{\left[a_{i}-\frac{x_{i}(b_{i}^{2}+b_{i}x_{2})}{b_{i}^{2}+b_{i}x_{3}+x_{4}}\right]}^{2}$	4	[−5, 5]	0.0003
F15	$4x_{1}^{2}-2.1x_{1}^{4}+\frac{1}{3}x_{1}^{6}+x_{1}x_{2}-4x_{2}^{2}+4x_{2}^{4}$	2	[−5, 5]	−1.0316
F16	$\sum_{i=1}^{n-1}\left[100{\left(x_{i+1}-x_{i}^{2}\right)}^{2}+{\left(x_{i}-1\right)}^{2}\right]$	2	[−5, 10]	0.398
F17	$\begin{array}{l} {}[1+{\left(x_{1}+x_{2}+1\right)}^{2}(19-14x_{1}+3x_{1}^{2}-14x_{2}+6x_{1}x_{2}+3x_{2}^{2})]\times \\ \left[30+{\left(2x_{1}-3x_{2}\right)}^{2}\times (18-32x_{1}+12x_{1}^{2}+48x_{2}-36x_{1}x_{2}+27x_{2}^{2})\right] \end{array}$	2	[−2, 2]	3
F18	$-\sum_{i=1}^{4}c_{i}\exp[-\sum_{j=1}^{4}a_{ij}{\left(x_{i}-p_{ij}\right)}^{2}]$	4	[0, 1]	−3.86
F19	$-\sum_{i=1}^{4}c_{i}\exp[-\sum_{j=1}^{6}a_{ij}{\left(x_{i}-p_{ij}\right)}^{2}]$	6	[0, 1]	−3.32
F20	$-{\sum_{i=1}^{5}\left[(x-a_{i}){\left(x-a_{i}\right)}^{T}+c_{i}\right]}^{-1}$	4	[0, 10]	−10.2
F21	$-{\sum_{i=1}^{7}\left[(x-a_{i}){\left(x-a_{i}\right)}^{T}+c_{i}\right]}^{-1}$	4	[0, 10]	−10.4
F22	$-{\sum_{i=1}^{10}\left[(x-a_{i}){\left(x-a_{i}\right)}^{T}+c_{i}\right]}^{-1}$	4	[0, 10]	−10.5

**Table 2 biomimetics-09-00298-t002:** Comparison of algorithm experimental results (30 dimensions).

Function	Algorithm	Mean	Std	Best	Worst
F1	PSO	2.41 × 10^5^	2.78 × 10^5^	1.92 × 10^5^	3.08 × 10^5^
	WDO	7.46 × 10^−19^	1.56 × 10^−18^	0	7.27 × 10^−18^
	GWO	3.57 × 10^−33^	5.73 × 10^−33^	0	2.62 × 10^−32^
	SCA	2.90 × 10^0^	3.73 × 10^0^	2.82 × 10^−3^	1.47 × 10^1^
	TSA	1.49 × 10^−12^	6.9 × 10^−13^	6.41 × 10^−13^	3.12 × 10^−12^
	DO	1.67 × 10^−6^	9.81 × 10^−7^	4.09 × 10^−7^	4.62 × 10^−6^
	PSODO	2.41 × 10^4^	2.78 × 10^3^	1.92 × 10^4^	3.08 × 10^4^
F2	PSO	4.17 × 10^2^	6.89 × 10^2^	6.26 × 10^2^	6.26 × 10^2^
	WDO	3.76 × 10^−10^	5.22 × 10^−10^	0	2.59 × 10^−9^
	GWO	7.14 × 10^−20^	5.09 × 10^−20^	0	2.06 × 10^−19^
	SCA	1.17 × 10^−2^	1.41 × 10^−2^	1.69 × 10^−4^	5.03 × 10^−2^
	TSA	5.98 × 10^−10^	1.55 × 10^−10^	2.95 × 10^−10^	1.07 × 10^−9^
	DO	5.38 × 10^−4^	2.42 × 10^−4^	1.70 × 10^−4^	1.26 × 10^−3^
	PSODO	2.53 × 10^−1^	3.35 × 10^−1^	0	1.27 × 10^0^
F3	PSO	5.68 × 10^4^	8.49 × 10^3^	3.08 × 10^4^	7.43 × 10^4^
	WDO	1.98 × 10^−14^	5.56 × 10^−14^	0	2.97 × 10^−13^
	GWO	2.54 × 10^−7^	9.27 × 10^−7^	2.44 × 10^−11^	5.06 × 10^−6^
	SCA	6.18 × 10^3^	4.51 × 10^4^	2.99 × 10^2^	1.66 × 10^4^
	TSA	1.62 × 10^4^	2.87 × 10^3^	1.02 × 10^4^	2.29 × 10^4^
	DO	2.13 × 10^1^	1.62 × 10^1^	4.28 × 10^0^	7.05 × 10^1^
	PSODO	2.83 × 10^−3^	8.06 × 10^−3^	0	3.83 × 10^−2^
F4	PSO	7.40 × 10^1^	4.85 × 10^0^	6.20 × 10^1^	8.06 × 10^1^
	WDO	4.97 × 10^−9^	1.52 × 10^−8^	8.41 × 10^−11^	8.35 × 10^−8^
	GWO	2.07 × 10^−8^	2.31 × 10^−8^	2.25 × 10^−9^	9.97 × 10^−8^
	SCA	2.93 × 10^1^	9.64 × 10^0^	1.42 × 10^1^	5.44 × 10^1^
	TSA	1.95 × 10^1^	2.94 × 10^0^	1.47 × 10^1^	2.66 × 10^1^
	DO	9.23 × 10^−1^	6.71 × 10^−1^	1.9 × 10^−1^	3.56 × 10^0^
	PSODO	3.3 × 10^−13^	4.56 × 10^−13^	0	1.7 × 10^−12^
F5	PSO	4.99 × 10^7^	1.27 × 10^7^	2.18 × 10^7^	7.55 × 10^7^
	WDO	2.86 × 10^1^	3.92 × 10^−2^	2.85 × 10^1^	2.87 × 10^1^
	GWO	2.67 × 10^1^	7.44 × 10^−1^	2.55 × 10^1^	2.87 × 10^1^
	SCA	1.40 × 10^4^	4.01 × 10^4^	4.23 × 10^1^	2.15 × 10^5^
	TSA	3.02 × 10^1^	1.59 × 10^1^	2.58 × 10^1^	1.13 × 10^2^
	DO	3.53 × 10^1^	2.50 × 10^1^	2.48 × 10^1^	1.35 × 10^2^
	PSODO	4.84 × 10^1^	3.77 × 10^1^	2.89 × 10^1^	1.80 × 10^2^
F6	PSO	9.10 × 10^4^	1.28 × 10^4^	5.76 × 10^4^	1.10 × 10^4^
	WDO	6.62 × 10^−2^	2.45 × 10^−2^	3.00 × 10^−2^	1.21 × 10^−1^
	GWO	4.72 × 10^−1^	2.77 × 10^−2^	6.11 × 10^−5^	1.00 × 10^0^
	SCA	6.82E × 10^0^	2.32 × 10^0^	4.62 × 10^0^	1.40 × 10^1^
	TSA	1.33 × 10^−12^	5.73 × 10^−13^	4.99 × 10^−13^	2.56 × 10^−12^
	DO	5.63 × 10^−6^	2.79 × 10^−6^	1.21 × 10^−6^	1.13 × 10^−5^
	PSODO	1.14 × 10^1^	1.01 × 10^1^	5.92 × 10^0^	6.24 × 10^1^
F7	PSO	7.40 × 10^0^	2.14 × 10^0^	3.84 × 10^0^	1.42 × 10^0^
	WDO	1.66 × 10^−4^	1.18 × 10^−4^	2.29× 10^−5^	4.98× 10^−4^
	GWO	1.22 × 10^−3^	8.31 × 10^−4^	3.32× 10^−4^	3.59× 10^−3^
	SCA	7.09 × 10^−2^	5.74 × 10^−1^	9.94× 10^−3^	2.29× 10^−1^
	TSA	2.50 × 10^−2^	6.36 × 10^−3^	1.33 × 10^−2^	3.91 × 10^−2^
	DO	1.77 × 10^−2^	8.81 × 10^−3^	4.37× 10^−3^	3.69 × 10^−2^
	PSODO	3.13 × 10^−2^	4.06 × 10^−2^	0	1.93× 10^−1^
F8	PSO	3.38 × 10^2^	2.11 × 10^1^	3.03 × 10^2^	3.87 × 10^2^
	WDO	5.61 × 10^1^	2.22 × 10^1^	1.94 × 10^1^	1.07 × 10^2^
	GWO	2.32 × 10^0^	3.27 × 10^0^	0	1.17 × 10^1^
	SCA	3.54 × 10^1^	2.72 × 10^1^	4.20 × 10^−2^	9.29 × 10^1^
	TSA	1.74 × 10^2^	2.30 × 10^1^	1.17 × 10^2^	2.04 × 10^2^
	DO	3.65 × 10^1^	1.93 × 10^1^	1.11 × 10^1^	8.38 × 10^1^
	PSODO	8.37× 10^−1^	1.73 × 10^0^	0	7.24 × 10^0^
F9	PSO	1.50 × 10^1^	5.38× 10^−1^	1.36 × 10^1^	1.59 × 10^1^
	WDO	2.35 × 10^−10^	3.27× 10^−10^	1.03 × 10^−11^	1.50 × 10^−9^
	GWO	4.45 × 10^−14^	5.49 × 10^−15^	3.95 × 10^−14^	6.08 × 10^−14^
	SCA	1.32 × 10^1^	9.47 × 10^0^	4.18 × 10^−2^	2.03 × 10^1^
	TSA	4.62 × 10^−7^	1.21 × 10^−7^	2.33 × 10^−7^	6.97 × 10^−7^
	DO	2.64 × 10^−4^	9.38 × 10^−5^	1.09 × 10^−4^	4.76 × 10^−4^
	PSODO	3.25 × 10^−10^	5.67 × 10^−11^	0	2.28 × 10^−10^
F10	PSO	2.56 × 10^2^	4.80 × 10^1^	1.42 × 10^2^	3.50 × 10^2^
	WDO	1.15 × 10^−2^	2.77 × 10^−2^	0	9.18 × 10^−2^
	GWO	3.38 × 10^−3^	8.64 × 10^−3^	0	3.75 × 10^−2^
	SCA	7.52 × 10^−1^	2.99 × 10^−1^	1.84 × 10^−1^	1.16 × 10^0^
	TSA	2.34 × 10^−7^	9.29 × 10^−7^	7.27 × 10^−12^	4.63 × 10^−6^
	DO	1.37 × 10^−2^	1.62 × 10^−2^	3.88 × 10^−6^	5.93 × 10^−2^
	PSODO	5.67 × 10^−1^	4.84 × 10^−1^	**0**	1.51 × 10^0^
F11	PSO	1.60 × 10^8^	4.46 × 10^7^	7.51 × 10^7^	2.59 × 10^8^
	WDO	7.07 × 10^−2^	1.45 × 10^−1^	1.91 × 10^−3^	5.21 × 10^−1^
	GWO	3.11 × 10^−2^	2.01 × 10^−2^	6.55 × 10^−3^	9.47 × 10^−2^
	SCA	7.24 × 10^3^	3.11 × 10^4^	5.00 × 10^−1^	1.66 × 10^5^
	TSA	9.08 × 10^−6^	2.09 × 10^−5^	1.91 × 10^−7^	8.95 × 10^−5^
	DO	1.04 × 10^−2^	3.16 × 10^−2^	1.10 × 10^−7^	1.04 × 10^−1^
	PSODO	1.20 × 10^0^	3.05 × 10^−1^	5.29 × 10^−1^	1.66 × 10^0^
F12	PSO	3.81 × 10^8^	1.08 × 10^8^	1.63 × 10^8^	5.56 × 10^8^
	WDO	4.51 × 10^−1^	9.03 × 10^−1^	1.96 × 10^−2^	3.19 × 10^0^
	GWO	4.03 × 10^−1^	1.69 × 10^−1^	9.94 × 10^−2^	7.13 × 10^−1^
	SCA	4.76 × 10^4^	2.20 × 10^5^	2.42 × 10^0^	1.19 × 10^6^
	TSA	1.25 × 10^−6^	1.29 × 10^−6^	1.09 × 10^−7^	5.46 × 10^−6^
	DO	5.44 × 10^−6^	4.02 × 10^−6^	6.65 × 10^−7^	2.10 × 10^−5^
	PSODO	3.42 × 10^0^	4.73 × 10^−1^	3.0 × 10^0^	4.78 × 10^0^
F13	PSO	1.04 × 10^0^	1.67 × 10^−1^	1.00 × 10^0^	1.92 × 10^0^
	WDO	9.77 × 10^0^	5.97 × 10^0^	1.00 × 10^0^	2.29 × 10^1^
	GWO	5.40 × 10^0^	4.51 × 10^0^	1.00 × 10^0^	1.27 × 10^1^
	SCA	2.12 × 10^0^	1.9 × 10^0^	1.00 × 10^0^	1.08 × 10^1^
	TSA	9.98 × 10^−1^	0	1.00 × 10^0^	9.98 × 10^−1^
	DO	9.98 × 10^−1^	3.16 × 10^−15^	1.00 × 10^0^	9.98 × 10^−1^
	PSODO	1.00 × 10^1^	1.34 × 10^1^	1.00 × 10^0^	7.69 × 10^1^
F14	PSO	4.00 × 10^−3^	2.42 × 10^−3^	9.92 × 10^−4^	9.79 × 10^−3^
	WDO	3.13 × 10^−4^	2.60 × 10^−5^	3.00 × 10^−4^	4.51 × 10^−4^
	GWO	3.08 × 10^−4^	3.54 × 10^−8^	3.00 × 10^−4^	3.08 × 10^−4^
	SCA	5.61 × 10^−4^	3.63 × 10^−4^	3.00 × 10^−4^	1.80 × 10^−3^
	TSA	3.07 × 10^−4^	2.38 × 10^−19^	3.00 × 10^−4^	3.07 × 10^−4^
	DO	3.07 × 10^−4^	2.89 × 10^−8^	3.00 × 10^−4^	3.08 × 10^−4^
	PSODO	1.51 × 10^−3^	1.17 × 10^−3^	3.00 × 10^−4^	4.15 × 10^−3^
F15	PSO	−1.03 × 10^0^	1.41 × 10^−3^	−1.03 × 10^0^	−1.03 × 10^0^
	WDO	−1.03 × 10^0^	9.84 × 10^−6^	−1.03 × 10^0^	−1.03 × 10^0^
	GWO	−1.03 × 10^0^	6.76 × 10^−9^	−1.03 × 10^0^	−1.03 × 10^0^
	SCA	−1.03 × 10^0^	3.24 × 10^−5^	−1.03 × 10^0^	−1.03 × 10^0^
	TSA	−1.03 × 10^0^	6.78 × 10^−16^	−1.03 × 10^0^	−1.03 × 10^0^
	DO	−1.03 × 10^0^	3.31 × 10^−13^	−1.03 × 10^0^	−1.03 × 10^0^
	PSODO	−9.03 × 10^−1^	7.41 × 10^−3^	−1.03 × 10^0^	−7.70 × 10^−1^
F16	PSO	4.98 × 10^−1^	1.92 × 10^−1^	3.98 × 10^−1^	1.20 × 10^0^
	WDO	3.98 × 10^−1^	1.69 × 10^−4^	3.98 × 10^−1^	3.99 × 10^−1^
	GWO	3.98 × 10^−1^	8.63 × 10^−5^	3.98 × 10^−1^	3.98 × 10^−1^
	SCA	4.00 × 10^−1^	9.38 × 10^−4^	3.98 × 10^−1^	4.02 × 10^−1^
	TSA	3.98 × 10^−1^	0	3.98 × 10^−1^	3.98 × 10^−1^
	DO	3.98 × 10^−1^	2.45 × 10^−11^	3.98 × 10^−1^	3.98 × 10^−1^
	PSODO	6.99 × 10^−1^	5.25 × 10^−1^	3.98 × 10^−1^	3.16 × 10^0^
F17	PSO	3.00 × 10^0^	3.29 × 10^−3^	3.00 × 10^0^	3.01 × 10^0^
	WDO	3.00 × 10^0^	1.05 × 10^−3^	3.00 × 10^0^	3.00 × 10^0^
	GWO	3.00 × 10^0^	1.71 × 10^−5^	3.00 × 10^0^	3.00 × 10^0^
	SCA	3.00 × 10^0^	2.90 × 10^−5^	3.00 × 10^0^	3.00 × 10^0^
	TSA	3.00 × 10^0^	1.86 × 10^−15^	3.00 × 10^0^	3.00 × 10^0^
	DO	3.00 × 10^0^	3.47 × 10^−9^	3.00 × 10^0^	3.00 × 10^0^
	PSODO	1.06 × 10^1^	1.19 × 10^1^	3.00 × 10^0^	4.22 × 10^1^
F18	PSO	−2.04 × 10^0^	6.47 × 10^−1^	−3.62 × 10^0^	−7.32 × 10^−1^
	WDO	−3.78 × 10^0^	2.35 × 10^−1^	−3.86 × 10^0^	−3.09 × 10^0^
	GWO	−3.77 × 10^0^	5.22 × 10^−1^	−3.86 × 10^0^	−1.00 × 10^0^
	SCA	−3.64 × 10^0^	7.54 × 10^−1^	−3.86 × 10^0^	−8.70 × 10^−1^
	TSA	−3.77 × 10^0^	5.23 × 10^−1^	−3.86 × 10^0^	−1.00 × 10^0^
	DO	−3.86 × 10^0^	3.23 × 10^−5^	−3.86 × 10^0^	−3.86 × 10^0^
	PSODO	−2.69 × 10^0^	9.60 × 10^−1^	−3.86 × 10^0^	−2.76 × 10^0^
F19	PSO	−1.76 × 10^0^	5.26 × 10^−1^	−2.80 × 10^0^	−7.89 × 10^−1^
	WDO	−3.16 × 10^0^	1.10 × 10^−1^	−3.32 × 10^0^	−3.02 × 10^0^
	GWO	−3.25 × 10^0^	7.78 × 10^−2^	−3.32 × 10^0^	−3.13 × 10^0^
	SCA	−3.02 × 10^0^	1.43 × 10^−1^	−3.27 × 10^0^	−2.60 × 10^0^
	TSA	−3.32 × 10^0^	1.33 × 10^−15^	−3.32 × 10^0^	−3.32 × 10^0^
	DO	−3.26 × 10^0^	6.03 × 10^−2^	−3.32 × 10^0^	−3.20 × 10^0^
	PSODO	−2.34 × 10^0^	5.04 × 10^−1^	−3.32 × 10^0^	−1.28 × 10^0^
F20	PSO	−5.65 × 10^−1^	2.31 × 10^−1^	−1.48 × 10^0^	−3.33 × 10^−1^
	WDO	−7.44 × 10^0^	3.03 × 10^0^	−1.02 × 10^1^	−2.54 × 10^0^
	GWO	−9.48 × 10^0^	1.75 × 10^0^	−1.02 × 10^1^	−5.06 × 10^0^
	SCA	−1.86 × 10^0^	1.57 × 10^0^	−5.02 × 10^0^	−4.97 × 10^−1^
	TSA	−1.02 × 10^1^	3.56 × 10^−8^	−1.02 × 10^1^	−1.02 × 10^1^
	DO	−5.97 × 10^0^	3.37 × 10^0^	−1.02 × 10^1^	−2.63 × 10^0^
	PSODO	−2.42 × 10^0^	1.32 × 10^0^	−1.02 × 10^1^	−1.04 × 10^0^
F21	PSO	−7.13 × 10^−1^	3.77 × 10^−1^	−2.46 × 10^0^	−4.24 × 10^−1^
	WDO	−8.27 × 10^0^	3.32 × 10^0^	−1.04 × 10^1^	−2.71 × 10^0^
	GWO	−1.04 × 10^1^	1.55 × 10^−4^	−1.04 × 10^1^	−1.04 × 10^1^
	SCA	−4.12 × 10^0^	1.53 × 10^0^	−7.40 × 10^0^	−9.06 × 10^−1^
	TSA	−1.04 × 10^1^	1.55 × 10^−15^	−1.04 × 10^1^	−1.04 × 10^1^
	DO	−6.63 × 10^0^	3.66 × 10^0^	−1.04 × 10^1^	−1.84 × 10^0^
	PSODO	−2.62 × 10^0^	1.42 × 10^0^	−1.04 × 10^1^	−1.09 × 10^0^
F22	PSO	−1.03 × 10^0^	3.31 × 10^−1^	−1.94 × 10^0^	−5.61 × 10^−1^
	WDO	−8.74 × 10^0^	3.08 × 10^0^	−1.05 × 10^1^	−1.66 × 10^0^
	GWO	−1.03 × 10^1^	1.48 × 10^0^	−1.05 × 10^1^	−2.42 × 10^0^
	SCA	−4.78 × 10^0^	1.38 × 10^0^	−9.10 × 10^0^	−2.72 × 10^0^
	TSA	−1.05 × 10^1^	1.75 × 10^−15^	−1.05 × 10^1^	−1.05 × 10^1^
	DO	−6.11 × 10^0^	3.77 × 10^0^	−1.05 × 10^1^	−1.68 × 10^0^
	PSODO	−2.87 × 10^0^	1.62 × 10^0^	−1.05 × 10^1^	−1.19 × 10^0^

**Table 3 biomimetics-09-00298-t003:** Comparison of algorithm experimental results (60 dimensions).

Function	Algorithm	Mean	Std	Best	Worst
F1	PSO	1.27 × 10^5^	8.79 × 10^3^	1.02 × 10^5^	1.42 × 10^5^
	WDO	1.07 × 10^−16^	3.47 × 10^−16^	6.20 × 10^−21^	1.70 × 10^−15^
	GWO	3.31 × 10^−21^	3.35 × 10^−21^	2.65 × 10^−22^	1.76 × 10^−20^
	SCA	1.29 × 10^3^	1.31 × 10^3^	4.11 × 10^1^	4.36 × 10^3^
	TSA	1.9 × 10^2^	4.12 × 10^1^	1.16 × 10^2^	2.87 × 10^2^
	DO	6.99 × 10^−3^	3.75 × 10^−3^	2.14 × 10^−3^	1.75 × 10^−2^
	PSODO	4.93 × 10^0^	1.83 × 10^1^	1.41 × 10^−8^	9.79 × 10^1^
F2	PSO	2.41 × 10^2^	1.37 × 10^1^	2.13 × 10^2^	2.71 × 10^2^
	WDO	3.18 × 10^−9^	4.18 × 10^−9^	5.66 × 10^−11^	1.74 × 10^−8^
	GWO	6.14 × 10^−13^	3.26 × 10^−13^	1.64 × 10^−13^	1.35 × 10^−12^
	SCA	8.71 × 10^−1^	1.12 × 10^0^	4.65 × 10^−2^	5.83 × 10^0^
	TSA	3.41 × 10^0^	9.89 × 10^−1^	1.95 × 10^0^	5.97 × 10^0^
	DO	3.93 × 10^−2^	9.78 × 10^−3^	2.04 × 10^−2^	6.04 × 10^−2^
	PSODO	7.26 × 10^−1^	8.46 × 10^−1^	5.59 × 10^−4^	3.21 × 10^0^
F3	PSO	2.08 × 10^5^	2.57 × 10^4^	1.64 × 10^5^	2.66 × 10^5^
	WDO	5.44 × 10^−12^	1.69 × 10^−11^	8.85 × 10^−16^	9.19 × 10^−11^
	GWO	1.95 × 10^−1^	3.16 × 10^−1^	1.9 × 10^−3^	1.47 × 10^0^
	SCA	6.58 × 10^4^	1.64 × 10^4^	3.07 × 10^4^	1.01 × 10^5^
	TSA	1.29 × 10^5^	1.03 × 10^4^	1.02 × 10^5^	1.48 × 10^5^
	DO	3.64 × 10^3^	1.97 × 10^3^	9.36 × 10^2^	8.82 × 10^3^
	PSODO	7.77 × 10^1^	1.55 × 10^2^	4.75 × 10^−3^	5.35 × 10^2^
F4	PSO	9.24 × 10^1^	1.99 × 10^0^	8.76 × 10^1^	9.65 × 10^1^
	WDO	7.74 × 10^−9^	1.29 × 10^−8^	2.23 × 10^−11^	5.46 × 10^−8^
	GWO	3.92 × 10^−4^	3.20 × 10^−4^	8.40 × 10^−5^	1.42 × 10^−3^
	SCA	7.29 × 10^1^	6.57 × 10^0^	6.16 × 10^1^	8.42 × 10^1^
	TSA	8.98 × 10^1^	4.08 × 10^0^	7.56 × 10^1^	9.45 × 10^1^
	DO	2.24 × 10^1^	6.81 × 10^0^	8.70 × 10^0^	3.54 × 10^1^
	PSODO	4.12 × 10^−1^	4.61 × 10^−1^	8.88 × 10^−3^	1.87 × 10^0^
F5	PSO	5.32 × 10^8^	5.62 × 10^7^	4.23 × 10^8^	6.35 × 10^8^
	WDO	5.84 × 10^1^	5.72 × 10^−3^	5.84 × 10^1^	5.84 × 10^1^
	GWO	5.72 × 10^1^	9.30 × 10^−1^	5.52 × 10^1^	5.86 × 10^1^
	SCA	1.12 × 10^7^	1.05 × 10^7^	6.03 × 10^5^	4.37 × 10^7^
	TSA	5.79 × 10^6^	1.37 × 10^6^	3.35 × 10^6^	9.40 × 10^6^
	DO	1.82 × 10^2^	1.43 × 10^2^	5.33 × 10^1^	5.96 × 10^2^
	PSODO	1.33 × 10^2^	1.29 × 10^2^	5.90 × 10^1^	4.35 × 10^2^
F6	PSO	1.31 × 10^5^	6.51 × 10^3^	1.17 × 10^5^	1.42 × 10^5^
	WDO	2.20 × 10^−1^	6.25 × 10^−2^	8.58 × 10^−2^	3.59 × 10^−1^
	GWO	2.77 × 10^0^	5.23 × 10^−1^	2.00 × 10^0^	3.81 × 10^0^
	SCA	1.20 × 10^3^	1.23 × 10^−3^	7.19 × 10^1^	5.60 × 10^3^
	TSA	1.97 × 10^2^	3.80 × 10^1^	1.23 × 10^2^	3.10 × 10^2^
	DO	1.74 × 10^−3^	8.31 × 10^−4^	4.77 × 10^−4^	3.73 × 10^−3^
	PSODO	1.54 × 10^1^	3.03 × 10^0^	1.16 × 10^1^	3.03 × 10^1^
F7	PSO	4.77 × 10^−2^	4.83 × 10^1^	3.64 × 10^2^	5.77 × 10^2^
	WDO	1.97 × 10^−4^	1.31 × 10^−4^	1.49 × 10^−5^	5.77 × 10^−4^
	GWO	2.75 × 10^−3^	1.30 × 10^−3^	7.16 × 10^−4^	5.38 × 10^−3^
	SCA	5.56 × 10^0^	4.32 × 10^0^	4.04 × 10^−1^	1.63 × 10^1^
	TSA	3.72 × 10^0^	8.45 × 10^−1^	2.18 × 10^0^	5.40 × 10^0^
	DO	9.66 × 10^−2^	3.25 × 10^−2^	4.34 × 10^−2^	1.64 × 10^−1^
	PSODO	4.46 × 10^−2^	7.77 × 10^−2^	2.67 × 10^−4^	3.95 × 10^−1^
F8	PSO	8.24 × 10^2^	3.26 × 10^1^	7.62 × 10^2^	8.81 × 10^2^
	WDO	1.57 × 10^2^	4.02 × 10^1^	1.05 × 10^2^	2.47 × 10^2^
	GWO	3.31 × 10^0^	4.01 × 10^0^	5.68 × 10^−13^	1.56 × 10^1^
	SCA	1.16 × 10^2^	8.26 × 10^1^	8.76 × 10^−1^	3.75 × 10^2^
	TSA	5.72 × 10^2^	3.43 × 10^1^	4.73 × 10^2^	6.26 × 10^2^
	DO	1.39 × 10^2^	6.11 × 10^1^	4.84 × 10^1^	2.70 × 10^2^
	PSODO	6.15 × 10^0^	1.73 × 10^1^	1.34 × 10^−5^	8.98 × 10^1^
F9	PSO	1.50 × 10^1^	5.38 × 10^−1^	1.36 × 10^1^	1.59 × 10^1^
	WDO	2.35 × 10^−10^	3.27 × 10^−10^	1.03 × 10^−11^	1.50 × 10^−9^
	GWO	4.45 × 10^−14^	5.49 × 10^−15^	3.95 × 10^−14^	6.08 × 10^−14^
	SCA	1.32 × 10^1^	9.47 × 10^0^	4.18 × 10^−2^	2.03 × 10^1^
	TSA	4.62 × 10^−7^	1.21 × 10^−7^	2.33 × 10^−7^	6.97 × 10^−7^
	DO	2.64 × 10^−4^	9.38 × 10^−5^	1.09 × 10^−4^	4.76 × 10^−4^
	PSODO	3.25 × 10^−1^	5.67 × 10^−1^	8.31 × 10^−5^	2.28 × 10^0^
F10	PSO	3.49 × 10^2^	1.83 × 10^2^	4.49 × 10^0^	7.22 × 10^2^
	WDO	9.50 × 10^−3^	3.00 × 10^−2^	9.79 × 10^−3^	1.27 × 10^−1^
	GWO	2.47 × 10^−3^	7.37 × 10^−3^	2.48 × 10^−3^	2.35 × 10^−1^
	SCA	1.62 × 10^1^	1.48 × 10^1^	1.13 × 10^0^	5.33 × 10^1^
	TSA	2.61 × 10^0^	2.83 × 10^−1^	1.99 × 10^0^	3.08 × 10^0^
	DO	1.65 × 10^−2^	1.03 × 10^−2^	4.37 × 10^−3^	4.43 × 10^−2^
	PSODO	5.13 × 10^−1^	5.45 × 10^−1^	0	1.31 × 10^0^
F11	PSO	9.13 × 10^8^	1.56 × 10^8^	5.82 × 10^8^	1.24 × 10^9^
	WDO	7.61 × 10^−3^	4.27 × 10^−3^	2.26 × 10^−3^	2.05 × 10^−2^
	GWO	1.02 × 10^−1^	5.67 × 10^−2^	3.54 × 10^−2^	3.41 × 10^−1^
	SCA	2.84 × 10^7^	3.18 × 10^7^	1.48 × 10^6^	1.54 × 10^8^
	TSA	2.04 × 10^7^	1.03 × 10^7^	6.37 × 10^6^	5.02 × 10^7^
	DO	1.63 × 10^0^	1.25 × 10^0^	6.26 × 10^−4^	3.96 × 10^0^
	PSODO	1.35 × 10^0^	1.35 × 10^−1^	0	1.74 × 10^0^
F12	PSO	1.91 × 10^9^	3.23 × 10^8^	1.27 × 10^9^	2.55 × 10^9^
	WDO	7.10 × 10^−1^	1.66 × 10^0^	9.52 × 10^−2^	6.99 × 10^0^
	GWO	2.46 × 10^0^	3.42 × 10^−1^	1.75 × 10^0^	3.02 × 10^0^
	SCA	6.63 × 10^7^	5.83 × 10^7^	3.45 × 10^5^	2.59 × 10^8^
	TSA	3.57 × 10^7^	1.15 × 10^7^	1.62 × 10^7^	6.85 × 10^7^
	DO	9.72 × 10^0^	1.42 × 10^1^	2.21 × 10^−3^	5.71 × 10^1^
	PSODO	7.15 × 10^0^	1.31 × 10^0^	6.00 × 10^0^	9.95 × 10^0^
F13	PSO	1.04 × 10^0^	1.67 × 10^−1^	1.00 × 10^0^	1.92 × 10^0^
	WDO	9.77 × 10^0^	5.97 × 10^0^	1.00 × 10^0^	2.29 × 10^1^
	GWO	5.40 × 10^0^	4.51 × 10^0^	1.00 × 10^0^	1.27 × 10^1^
	SCA	2.12 × 10^0^	1.9 × 10^0^	1.00 × 10^0^	1.08 × 10^1^
	TSA	9.98 × 10^−1^	0	1.00 × 10^0^	9.98 × 10^−1^
	DO	9.98 × 10^−1^	3.16 × 10^−15^	1.00 × 10^0^	9.98 × 10^−1^
	PSODO	1.00 × 10^1^	1.34 × 10^1^	1.00 × 10^0^	7.69 × 10^1^
F14	PSO	5.42 × 10^−3^	7.72 × 10^−3^	3.00 × 10^−4^	2.04 × 10^−2^
	WDO	5.43 × 10^−4^	4.71 × 10^−4^	3.00 × 10^−4^	1.60 × 10^−3^
	GWO	3.02 × 10^−3^	6.92 × 10^−3^	3.00 × 10^−4^	2.04 × 10^−2^
	SCA	1.02 × 10^−3^	3.52 × 10^−4^	3.00 × 10^−4^	1.54 × 10^−3^
	TSA	3.32 × 10^−4^	3.11 × 10^−5^	3.00 × 10^−4^	4.35 × 10^−3^
	DO	5.90 × 10^−4^	3.04 × 10^−4^	3.00 × 10^−4^	1.06 × 10^−3^
	PSODO	4.25 × 10^−3^	4.20 × 10^−3^	3.00 × 10^−4^	1.43 × 10^−2^
F15	PSO	−1.03 × 10^0^	1.42 × 10^−3^	−1.03 × 10^0^	−1.02 × 10^0^
	WDO	−1.03 × 10^0^	9.36 × 10^−6^	−1.03 × 10^0^	−1.03 × 10^0^
	GWO	−1.03 × 10^0^	7.73 × 10^−9^	−1.03 × 10^0^	−1.03 × 10^0^
	SCA	−1.03 × 10^0^	3.54 × 10^−5^	−1.03 × 10^0^	−1.03 × 10^0^
	TSA	−1.03 × 10^0^	6.78 × 10^−16^	−1.03 × 10^0^	−1.03 × 10^0^
	DO	−1.03 × 10^0^	1.10 × 10^−13^	−1.03 × 10^0^	−1.03 × 10^0^
	PSODO	−9.87 × 10^−1^	4.85 × 10^−2^	−1.03 × 10^0^	−8.67 × 10^−1^
F16	PSO	3.99 × 10^−1^	9.44 × 10^−4^	3.98 × 10^−1^	4.02 × 10^−1^
	WDO	3.98 × 10^−1^	1.98 × 10^−4^	3.98 × 10^−1^	3.99 × 10^−1^
	GWO	3.98 × 10^−1^	3.83 × 10^−7^	3.98 × 10^−1^	3.98 × 10^−1^
	SCA	4.00 × 10^−1^	2.72 × 10^−3^	3.98 × 10^−1^	4.12 × 10^−1^
	TSA	3.98 × 10^−1^	0	3.98 × 10^−1^	3.98 × 10^−1^
	DO	3.98 × 10^−1^	8.81 × 10^−11^	3.98 × 10^−1^	3.98 × 10^−1^
	PSODO	6.22 × 10^−1^	2.79 × 10^−1^	3.98 × 10^−1^	1.42 × 10^0^
F17	PSO	3.02 × 10^0^	2.00 × 10^−2^	3.00 × 10^0^	3.09 × 10^0^
	WDO	3.00 × 10^0^	1.13 × 10^−3^	3.00 × 10^0^	3.01 × 10^0^
	GWO	3.00 × 10^0^	1.17 × 10^−5^	3.00 × 10^0^	3.00 × 10^0^
	SCA	3.00 × 10^0^	3.02 × 10^−5^	3.00 × 10^0^	3.00 × 10^0^
	TSA	3.00 × 10^0^	1.80 × 10^−15^	3.00 × 10^0^	3.00 × 10^0^
	DO	3.00 × 10^0^	4.43 × 10^−9^	3.00 × 10^0^	3.00 × 10^0^
	PSODO	6.68 × 10^0^	4.07 × 10^0^	3.00 × 10^0^	1.9 × 10^1^
F18	PSO	−2.71 × 10^0^	8.96 × 10^−1^	−3.86 × 10^0^	−8.35 × 10^−1^
	WDO	−3.62 × 10^0^	7.38 × 10^−1^	−3.86 × 10^0^	−1.00 × 10^0^
	GWO	−3.47 × 10^0^	9.89 × 10^−1^	−3.86 × 10^0^	−1.00 × 10^0^
	SCA	−3.84 × 10^0^	1.22 × 10^−2^	−3.86 × 10^0^	−3.81 × 10^0^
	TSA	−3.78 × 10^0^	2.36 × 10^−1^	−3.86 × 10^0^	−3.09 × 10^0^
	DO	−3.86 × 10^0^	2.32 × 10^−5^	−3.86 × 10^0^	−3.86 × 10^0^
	PSODO	−2.65 × 10^0^	9.52 × 10^−1^	−3.86 × 10^0^	−5.02 × 10^−1^
F19	PSO	−1.76 × 10^0^	5.26 × 10^−1^	−2.80 × 10^0^	−7.89 × 10^−1^
	WDO	−3.16 × 10^0^	1.10 × 10^−1^	−3.32 × 10^0^	−3.02 × 10^0^
	GWO	−3.25 × 10^0^	7.78 × 10^−2^	−3.32 × 10^0^	−3.13 × 10^0^
	SCA	−3.02 × 10^0^	1.43 × 10^−1^	−3.27 × 10^0^	−2.60 × 10^0^
	TSA	−3.32 × 10^0^	1.33 × 10^−15^	−3.32 × 10^0^	−3.32 × 10^0^
	DO	−3.26 × 10^0^	6.03 × 10^−2^	−3.32 × 10^0^	−3.20 × 10^0^
	PSODO	−2.34 × 10^0^	5.04 × 10^−1^	−3.32 × 10^0^	−1.28 × 10^0^
F20	PSO	−7.07 × 10^−1^	3.72 × 10^−1^	−1.88 × 10^0^	−3.00 × 10^−1^
	WDO	−7.45 × 10^0^	3.02 × 10^0^	−1.02 × 10^1^	−2.59 × 10^0^
	GWO	−8.97 × 10^0^	2.17 × 10^0^	−1.02 × 10^1^	−5.10 × 10^0^
	SCA	−3.18 × 10^0^	1.92 × 10^0^	−5.74 × 10^0^	−8.78 × 10^−1^
	TSA	−1.02 × 10^1^	7.01 × 10^−15^	−1.02 × 10^1^	−1.02 × 10^1^
	DO	−6.98 × 10^0^	3.54 × 10^0^	−1.02 × 10^1^	−2.63 × 10^0^
	PSODO	−2.35 × 10^0^	1.46 × 10^0^	−1.02 × 10^1^	−1.03 × 10^0^
F21	PSO	−9.00 × 10^−1^	3.72 × 10^−1^	−2.18 × 10^0^	−3.64 × 10^−1^
	WDO	−9.00 × 10^0^	2.85 × 10^0^	−1.04 × 10^1^	−2.73 × 10^0^
	GWO	−1.02 × 10^1^	9.63 × 10^−1^	−1.04 × 10^1^	−5.13 × 10^0^
	SCA	−3.83 × 10^0^	1.97 × 10^0^	−7.49 × 10^0^	−9.07 × 10^−1^
	TSA	−1.04 × 10^1^	1.58 × 10^−15^	−1.04 × 10^1^	−1.04 × 10^1^
	DO	−6.88 × 10^0^	3.92 × 10^0^	−1.04 × 10^1^	−1.84 × 10^0^
	PSODO	−2.91 × 10^0^	1.73 × 10^0^	−1.04 × 10^1^	−1.28 × 10^0^
F22	PSO	−1.62 × 10^0^	5.92 × 10^−1^	−3.01 × 10^0^	−7.54 × 10^−1^
	WDO	−8.27 × 10^0^	3.33 × 10^0^	−1.05 × 10^1^	−2.65 × 10^0^
	GWO	−1.04 × 10^1^	2.44 × 10^−4^	−1.05 × 10^1^	−1.04 × 10^1^
	SCA	−4.66 × 10^0^	2.36 × 10^0^	−1.05 × 10^1^	−5.21 × 10^−1^
	TSA	−1.04 × 10^1^	1.51 × 10^−15^	−1.05 × 10^1^	−1.04 × 10^1^
	DO	−6.36 × 10^0^	3.49 × 10^0^	−1.05 × 10^1^	−1.83 × 10^0^
	PSODO	−2.60 × 10^0^	1.19 × 10^0^	−1.05 × 10^1^	−1.23 × 10^0^

**Table 4 biomimetics-09-00298-t004:** Comparison of 7 algorithms for three-bar truss design problems.

Algorithms	*x* _1_	*x* _2_	*f* _min_
PSO	0.7877	0.4108	267.867
GWO	0.7423	0.5583	263.9246
WDO	0.7981	0.4342	271.6039
SCA	0.7931	0.3956	264.4124
TSA	0.7891	0.4069	263.9383
DO	0.7788	0.4405	263.9052
PSODO	0.7886	0.4082	263.8959

**Table 5 biomimetics-09-00298-t005:** Comparison of 7 algorithms for three-bar truss design problems.

Algorithms	Best	Mean
PSO	267.8674	268.6879
GWO	263.9246	263.9354
WDO	271.6039	271.6119
SCA	264.4124	264.4182
TSA	263.9383	263.9481
DO	263.9052	263.9196
PSODO	263.8959	263.8989

**Table 6 biomimetics-09-00298-t006:** Comparison of pressure vessel design algorithms.

Algorithms	*T_s_*	*T_h_*	*R*	*L*	*f* _min_
PSO	14.1915	20.8044	43.5457	159.4919	11834.3516
GWO	13.1185	7.2311	42.0889	176.7684	6145.4546
WDO	17.9323	9.1698	57.1068	50.4563	6572.7595
SCA	14.2199	6.9543	45.3367	140.2538	5900.3202
TSA	13.6612	7.1734	45.1328	142.7379	7319.0007
DO	12.7342	7.0045	42.0823	178.2653	8131.006
PSODO	13.2089	7.4947	42.0984	176.6366	5885.3328

**Table 7 biomimetics-09-00298-t007:** Comparison of algorithms for pressure vessel design problems.

Algorithms	Best	Mean
PSO	11,834.3516	37,204.2840
GWO	6145.4546	10,708.9581
WDO	6572.7595	51,583.2265
SCA	5900.3202	9190.98310
TSA	7319.0007	24,343.9637
DO	8131.006	17,448.3396
PSODO	5885.3328	19,766.3177

**Table 8 biomimetics-09-00298-t008:** Comparison of compression spring design algorithms.

Algorithms	*d*	*D*	*N*	*f* _min_
PSO	0.0572	0.5056	5.9636	0.01631
GWO	0.0586	0.5488	5.1551	0.01319
WDO	0.0579	0.5277	5.5187	0.01333
SCA	0.0589	0.5568	5.0098	0.01354
TSA	0.0553	0.4512	7.3683	0.01296
DO	0.055	0.3171	14.1011	0.01289
PSODO	0.051	0.3175	14.0183	0.01271

## Data Availability

The original contributions presented in this study are included in this article; further inquiries can be directed to the corresponding author.

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
