# Peer review of "Solving Engineering Optimization Problems Based on Multi-Strategy Particle Swarm Optimization Hybrid Dandelion Optimization Algorithm"

_biomimetics, 2024, doi:10.3390/biomimetics9050298_

Round 1

Reviewer 1 Report

Comments and Suggestions for Authors

This research considers the combination of the Particle Swarm Optimization (PSO) and the Dandelion Optimization (DO). This is an interesting thought and would be a useful reference for peer researchers in both fields. Nonetheless, the current manuscript needs improvement to be suitable for publication.

[Major]

The results seem to be overwhelmingly better than the original DO (see Fig.2) but the results also seem to be single trials. Average of multiple runs would be more convincing. If it is true after investigating multiple trials, it would be nice to discuss the intuition of why it works so well. If it is not true, a discussion of caveats and their source would be helpful. In the current conclusion section, the authors write 'this algorithm only tests some engineering application problems, but does not test more complex mechanical engineering parameter optimization problems.' This is not a shortcoming of the algorithm. Instead it is a shortcoming of the current manuscript.

The normal distribution in step 4 (if I understand correctly) of the PSODO algorithm is not clear. It is possible the author means a unit normal distribution (mean=0, standard deviation =1). Can the author explain why this particular number is used?

[Minor]

The 'curve's in Fig.2 are unnecessary.

Fig.5: These curves sticking to each other are impossible to discern from each other. Stretching the x-axis and/or y-axis could be helpful.

Comments on the Quality of English Language

This manuscript needs further proofreading. Examples:

Author list: Yinggao Yue and what ?

`which is based on the problems of slow optimization speed and easy to fall into local extremum': the `easy' here has grammar issue.

`dandelion algorithm (i.e.; rising; falling and falling).': Do the authors really mean falling and falling?'

`have been widely concerned by scholars at home and abroad:' This is supposed to be an article in an international journal. There is no home per se.

Author Response

Dear Editor,

Thank you for allowing a resubmission of our manuscript, with an opportunity to address the reviewers’ comments.

We are uploading our point-by-point response to the comments (below) (response to reviewers).

Reviewer 2 Report

Comments and Suggestions for Authors

This manuscript presented a novel approach for solving the engineering optimization problems which was based on the hybridization of two classical algorithms. Please consider the following comments:

1. Please revise the entire abstract, please check the cases where ";" can be replaced with ",".

2. With respect to the Introduction section, please check if it is possible to add two paragraphs. The first one should present the main contributions of the article, using bullet points for each contribution. Then, the second one should describe how the paper is organized, for example: "The paper is organized as follows: Section 2 presents the related works, Section 3 presents …" and so on. 

3. Regarding page 6, where it is mentioned "r1 and r2 are random numbers" please describe the ranges of these two numbers.

4. Please revise the entire algorithm 1 (Table 1) for a better readability. The input and output should describe only the values which are used as inputs (POP, Dim T, …) and so on, while the output should present only the value which is returned. Also, please remove the extra line 10. Please also revise Table 2 from this perspective.

5. Please revise the last phrase from the Conclusions section for clarity: "At the same time, it is applied to other practical problems …". Also, please extend the Conclusions section with 3 possible future research directions. 

Author Response

(The authors gave the same response as above.)

Round 2

Reviewer 1 Report

Comments and Suggestions for Authors

I appreciate the authors' effort to address my comments and am satisfied with the changes.